# Single-cell transcriptomic profiling of *C. elegans* Q neuroblast lineage during migration and differentiation

Felipe L. Teixeira ⓘ*, Brian Sanderson ⓘ, Jennifer L. Hackett ⓘ, Erik A. Lundquist ⓘ*

Department of Molecular Biosciences, University of Kansas, Lawrence, Kansas, United States of America

* teixeirafl@ku.edu (FLT); erikl@ku.edu (EAL)

## Abstract

Proper migration and differentiation of neuroblasts into neurons are essential for the development of a healthy nervous system. In this context, the asymmetrical migration of *Caenorhabditis elegans* Q neuroblasts provides a powerful model for studying the genetic aspects of neuronal migration *in vivo* at single-cell resolution. We isolated Q lineage cells at various stages of development using FACS and employed single-cell RNA sequencing to investigate the molecular mechanisms underlying the migration and differentiation of these neuroblasts. We created a robust transcriptomic differentiation map of the Q neuroblast lineage and used established markers to identify each cell in the lineage. Our results revealed novel genes not previously described in these cells and linked the expression of known genes to specific stages of Q lineage progression. Furthermore, functional enrichment and imaging provided evidence that the parent Q cells are initially specified with an epithelial-like identity and undergo epithelial-mesenchymal transition during the early stages of migration. We also identified novel Wnt-related expression, including left-right asymmetric expression of *cwn-1* and *cwn-2*, and the involvement of the Wnt/β-catenin asymmetry pathway in the Q lineage. Our work offers a high-resolution view of neuroblast development, showcasing the power of single-cell transcriptomics to reveal stage-specific regulatory programs.

## Introduction

Migration of neurons and neuroblasts from their birthplace to their final position within the developing nervous system is a critical process, and it ensures the correct integration of these cells into functional neural circuits [1,2]. Improper migration is associated with a broad spectrum of developmental and neurological disorders [3–5]. Given the complexity of studying neuronal migration at single-cell resolution *in vivo*, we often need to rely on model systems to investigate the molecular mechanisms associated with the migratory process. One such model is the asymmetric migration

**Data availability statement:** Raw single-cell RNA sequencing data is publicly available through NCBI's Sequence Read Archive (SRA). BioProject Accession: PRJNA1300790 (https://www.ncbi.nlm.nih.gov/bioproject/PRJNA1300790).

**Funding:** This study was funded by the University of Kansas Center for Genomics and by the National Institutes of Health, specifically the National Institute of General Medical Sciences (grant numbers P30GM145499, P20GM113117, and P20GM103418) and National Institute of Neurological Disorders and Stroke (grant number R01NS115467). The funders had no role in study design, data collection and analysis, decision to publish, or preparation of the manuscript.

**Competing interests:** The authors have declared that no competing interests exist.

of Q neuroblasts during postembryonic development of the *Caenorhabditis elegans* nervous system [6], which enables detailed analysis of gene expression dynamics during neuronal development. This system allows the identification of candidate genes crucial for neuroblast migration and differentiation, contributing to our understanding of molecular mechanisms associated with nervous system development.

The Q cells are bilateral neuroblasts born embryonically approximately one hour before hatching between the seam cells V4 and V5 [7,8]. Following hatching, they undergo two phases of left-right (L-R) asymmetric migration (Fig 1A; reviewed in [9]). In the first one, the right (QR) and left (QL) side neuroblasts migrate anteriorly and posteriorly, respectively, to a position dorsal to their neighboring seam cells. The second phase of migration begins after they divide atop the seam cells into one anterior (Qx.a) and one posterior (Qx.p) daughter cell, each following different genetic programs as they continue migrating along the anterior-posterior axis. During its migration, Qx.a asymmetrically divides into Qx.aa, which undergoes programmed cell death, and Qx.ap, which ultimately gives rise to one sensory neuron in each side, AQR in the QR lineage and PQR in the QL lineage. Qx.p divides into Qx.pp, that is programmed to die, and Qx.pa, which will divide again to give rise to two neurons, the mechanosensory neurons AVM and PVM from QR.paa and QL.paa, respectively, and the interneurons SDQR and SDQL, from QR.pap and QL.pap, respectively (Fig 1B) [8,10–18].

The migration of Q descendants, but not of the parent Q cells, relies heavily on Wnt signaling and the Hox genes *mab-5* and *lin-39*. The Wnt ligand EGL-20 triggers the activation of the canonical Wnt/β-catenin signaling pathway in QL, which ultimately leads to upregulation of *mab-5* and downregulation of *lin-39* in the QL lineage. MAB-5 is necessary and sufficient for the posterior migration of QL descendants, acting by directly repressing *lin-39* expression. In the absence of MAB-5 in the QR lineage, *lin-39* remains expressed and promotes anterior migration [16,18–27].

In this work, we used single-cell RNA sequencing (scRNA-seq) to map the transcriptomic changes the Q neuroblast lineage undergoes as cells migrate, divide, and differentiate into neurons. While single-cell transcriptomics have been applied to *C. elegans* in multiple contexts [28–34], transcriptional data from the first few hours after hatching, when Q lineage cells initiate their developmental programs, have remained limited. We used known gene expression patterns and trajectory analysis to identify each cell in the QL and QR lineages. Data analysis revealed novel genes and pathways associated with Q lineage differentiation, including both established and previously unassociated genes. Early Q cells exhibited an epithelial-like profile, consistent with their birth from an epithelial seam cell, and an epithelial to mesenchymal transition (EMT) that accompanies initial Q cell migration. Later expression profiles delineate neuroblast fates (e.g., Qx.a, Qx.p, and Qx.pa) versus neuronal fates (e.g., Qx.ap, Qx.paa, and Qx.pap). Overall, our study gives a detailed look at neuroblast development and shows how useful single-cell transcriptomics is for identifying cell-specific gene expression during this process. Further analysis of genes from our dataset promises to uncover additional mechanisms governing neuroblast migration and differentiation, with potential relevance to the neurodevelopment of humans and other animals.

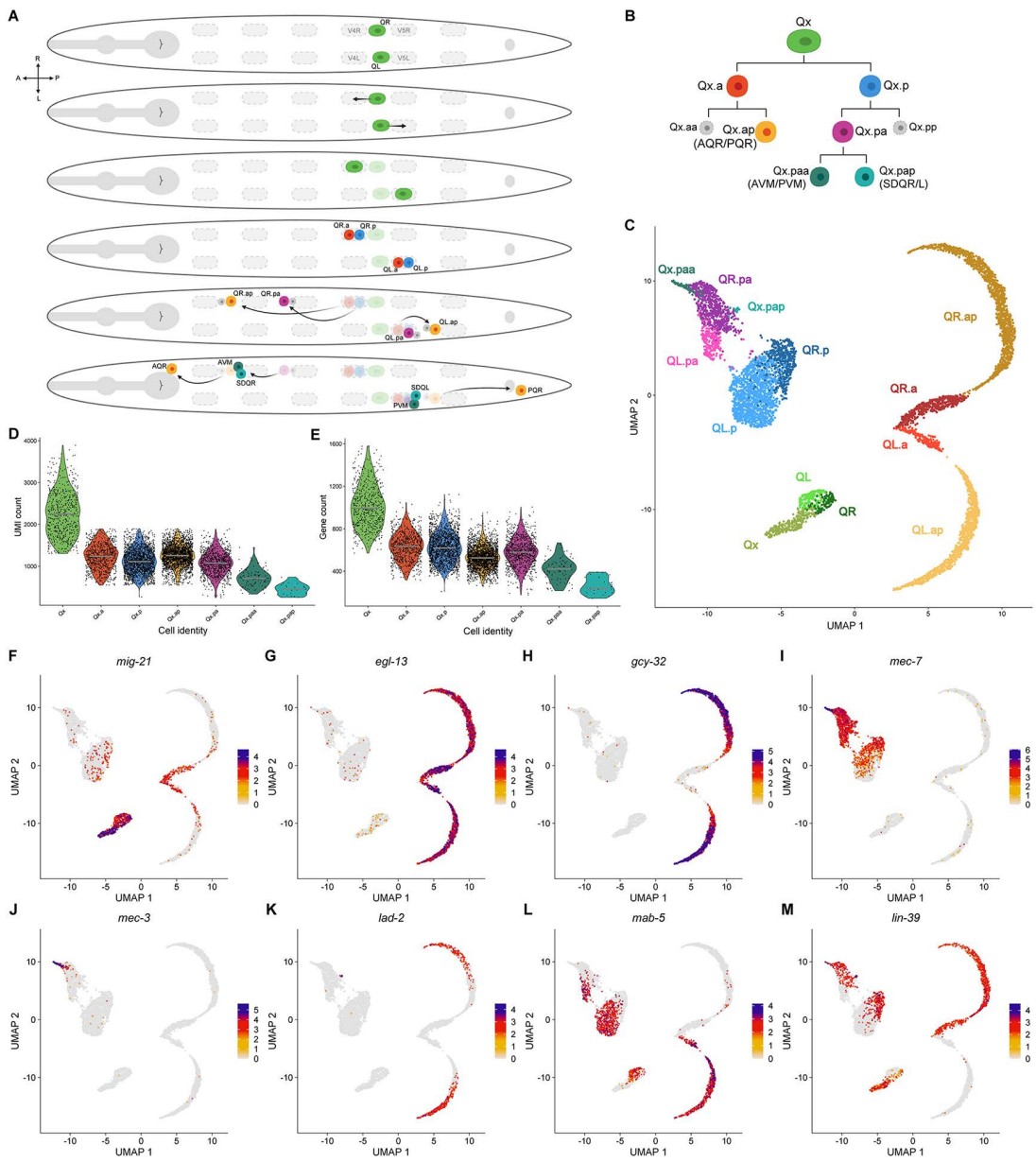

**Fig 1. The Q cell lineage migration and differentiation. (A)** Schematic representation of the migration and cell divisions of Q neuroblasts and their descendants. **(B)** The Q cell lineage. **(C)** UMAP plot of the Q cell lineage during development with each cell type identified. Cell stage boundaries (e.g., separation between QR.a and QR.ap) were inferred from transcriptional differences and may be slightly offset from the actual timing of cell division. **(D-E)** Violin plots showing the variation in number of UMIs (D) or genes expressed (E) across the different Q cell types. **(F-M)** UMAP plots showing the expression pattern of marker genes used to identify the different Q cell types.

## Results

### Transcriptomic map of the Q neuroblast lineage during early L1 development

We isolated Q cells from early, well-fed L1s to map transcriptomic changes associated with Q neuroblast lineage development using scRNA-seq (see Materials and Methods). We used two markers that are co-expressed exclusively in Q cells

(*Pegl-17::gfp*; *Pscm::mCherry*) as previously reported [18]. Fluorescence-activated cell sorting (FACS) was used to isolate the Q cells from early L1 populations spanning all stages of the Q lineage development. Following transcript quantification with CellRanger [35] and normalization and integration of three different, independent experiments with Seurat [36], we used the uniform manifold approximation and projection (UMAP) algorithm [37,38] to segregate cells into different groups (Fig 1C) and annotated each group by analyzing the expression of marker genes with known expression patterns in the Q cell lineage (Fig 1F-M), as described below. We identified the parent Q neuroblast and all Q cell descendants, with clear differentiation between left and right-side cells in most cell types.

Our dataset contained 6,743 single cells after quality control and Q lineage filtering, with a median of 1,216.6 unique molecular identifiers (UMIs) and 580 genes detected per cell. The number of UMIs and genes detected was not uniform across cell types, and the variation became especially pronounced after the first and third lineage divisions, but not after the second division. This pattern suggests that the mRNA content is more evenly distributed when both daughter cells are functional but becomes asymmetric when one daughter is programmed to die (Fig 1B, D, E; S1 Table). These biological differences in mRNA content, together with technical variability, made it difficult to apply uniform quality-control thresholds. To address this, we customized our processing for individual cell types, allowing us to confidently remove doublets and low-quality cells from our dataset without discarding healthy single cells.

We observed a variation between the number of cells on the left or right side within each cell type that may be attributed to challenges in isolating certain cell types and/or inherent difference in lineage progression between L-R. The final descendants of the posterior Q cell lineage, Qx.paa and Qx.pap, were the smallest populations in our dataset (S1Table), which prevented a reliable separation between L-R within the cluster. The low counts of these cells are most likely due to the developmental time window used to synchronize the initial L1 population, which limited enrichment of these later-stage cells. A potential contribution from the unexpected behavior of the sorting marker *Pegl-17::gfp* to cell variability, however, cannot be excluded. The expression of *egl-17*, a well-established marker of the Q cell lineage [39], was not consistent throughout the Q lineage. The high expression of *egl-17* in the Q neuroblasts is downregulated after the first cell division. Though it has a small upregulation in the anterior Q cell lineage, the expression in the posterior Q cell lineage is negligible (S1 Fig), decreasing the GFP signal with each cell division. A possible asymmetric distribution of the already low levels of GFP in Qx.pax may contribute to the fourfold difference in cell counts between Qx.paa and Qx.pap, but further analyses are needed to test this hypothesis.

In our raw data, we identified a cluster of cells with a high percentage of RNA counts attributed to mitochondrial genes, which typically indicates low cell viability [40]. Since post-sort check indicated high viability of sorted cells, we suspect that this cluster might represent Qx.aa and Qx.pp, the Q descendants programmed to die, rather than technical artifacts. However, we removed them from the dataset during downstream processing as their low transcript counts prevents reliable analysis.

Notably, Q cell development appeared robust across the two food sources. We observed no differences in clustering or cell-type composition attributable to hatching on OP50 versus PA14 (S1 Fig). Furthermore, differential gene expression analysis did not reveal any food-source-associated differences that could confound our results (S1 File).

## Expression of Q lineage marker genes used for cluster annotation

Several genes have been associated with the first stage of Q cell migration, including *mig-21*, which encodes a transmembrane protein that is required for the initial polarization and migration of these neuroblasts [13,14]. Expression of *mig-21* is transient in Q cells and restricted to the early migration stage, as evidenced by single molecule mRNA fluorescence *in situ* hybridization (smFISH) [13]. The higher expression of *mig-21* during early Q development allowed identification of the parent Q neuroblast in our dataset (Fig 1F). Clustering showed that these neuroblasts appear to go through two major cell states, regardless of left and right asymmetry.

Initial distinction between the anterior and posterior daughter lineages was achieved by analysis of *egl-13* expression pattern (Fig 1G). The transcription factor (TF) EGL-13 is a major regulator of Qx.a lineage differentiation and is necessary

for proper neuronal fate in Qx.ap cells [41]. In our dataset, high expression of *egl-13* defined the Qx.a and Qx.ap clusters. The expression pattern of several guanylate cyclases expressed by AQR and PQR, such as *gcy-32*, *gcy-35*, and *gcy-36* [42,43], contributed to confirm the Qx.a lineage cluster and to differentiate Qx.ap from Qx.a (Fig 1H; S1 Fig).

The expression of neuronal markers associated with Qx.p descendants confirmed identification of the clusters associated with the posterior daughter lineage. Since some of them were activated at different stages and remain active throughout development, analyzing the different expression patterns was enough to allow identification of each new descendant. For example, among genes associated with mechanosensory neurons such as AVM and PVM [10,29,34], the expression of *mec-7* started earlier, in Qx.p, while *mec-18* was expressed in late Qx.pa, and *mec-3* was expressed in Qx.paa (Fig 1I, J; S1 Fig). Although lowly expressed, the combined detection of markers associated with the SDQ neurons, such as *lad-2*, *ceh-31*, *ZK265.7*, and *ceh-43* [29,34,44,45], in the same cluster allowed identification of Qx.pap (Fig 1K; S1 Fig). Some markers associated with the final descendants of the posterior lineage, such as *mec-4*, however, were not detected, suggesting they are expressed in a later stage of Qx.pax development that we did not capture (S1 Fig).

The L-R asymmetry established by expression of *mab-5* or *lin-39* was clearly represented in our dataset (Fig 1L, M). Cells expressing either of these *hox* genes were easily distinguishable in most clusters, with two exceptions. We could not distinguish between QL and QR before *mab-5* was activated in QL, suggesting that initial asymmetry during the first stage of migration might be influenced by a non-autonomous source or that the effect of cell autonomous genes on the cell transcriptome is too subtle to be detected with our approach. Due to the low number of isolated cells in these clusters, we could not distinguish left from right in Qx.paa or Qx.pap cells either, though higher overall expression of *lin-39* suggests most of these cells might be associated with the right side.

## Trajectory inference and gene expression dynamics associated with Q lineage progression

We used Monocle 3 to investigate Q cell lineage progression, ordering cells along a trajectory based on their transition between different cell states [46–49]. The pseudotime analysis was rooted by temporal expression of *mig-21* [13] and measures the progression of cells through Q lineage development (Fig 2A). The trajectory starts in the Qx cells and progresses across pseudotime, without much differentiation between L-R, until branching out to follow the anterior and posterior descendants. While the trajectory in the anterior lineage can distinguish between L-R and follows the expected progression (from Qx.a to Qx.ap), it did not recognize the L-R asymmetry in the posterior lineage properly. Also, the low number of Qx.paa and Qx.pap cells caused a poor separation from their parent Qx.pa cell cluster, hindering the trajectory analysis at the third cell division.

We used the graph_test function in Monocle3 to assess gene expression variation between cell groups and the find_gene_modules function to group co-regulated genes into modules. We identified 25 modules of co-regulated genes during Q cell lineage development (Fig 2C; S2 File). Hierarchical clustering showed a greater distance between cell types than it did between L-R asymmetry within each cell type, following the trend of lower difference between L-R than lineage progression observed in the UMAP clustering patterns and trajectory inference. Also, the final descendants were clustered further apart from other cell types, showing that the changes in transcriptome associated with the differentiation into neurons differ substantially from the programming associated with migration and initial differentiation of the neuroblasts.

We divided the 25 modules into 4 major groups based on their hierarchical clustering: Group 1 was more associated with the final descendant neurons; group 2 was more associated with the anterior lineage neuroblasts; group 3 was more associated with the posterior lineage neuroblasts; and group 4 was more associated with the parent Q neuroblasts (S2 File). We used WormCat 2.0 [50] to functionally annotate the gene modules identified by Monocle3, providing an overview of the biological programs represented in each module group (Fig 2B; S2 File). Genes were assigned to functional categories based on their annotated physiological roles in *C. elegans*. The categories with most genes assigned to were metabolism, with 553 genes, and signaling, with 502 genes – the only category present in all 4 groups. Group 1 was the

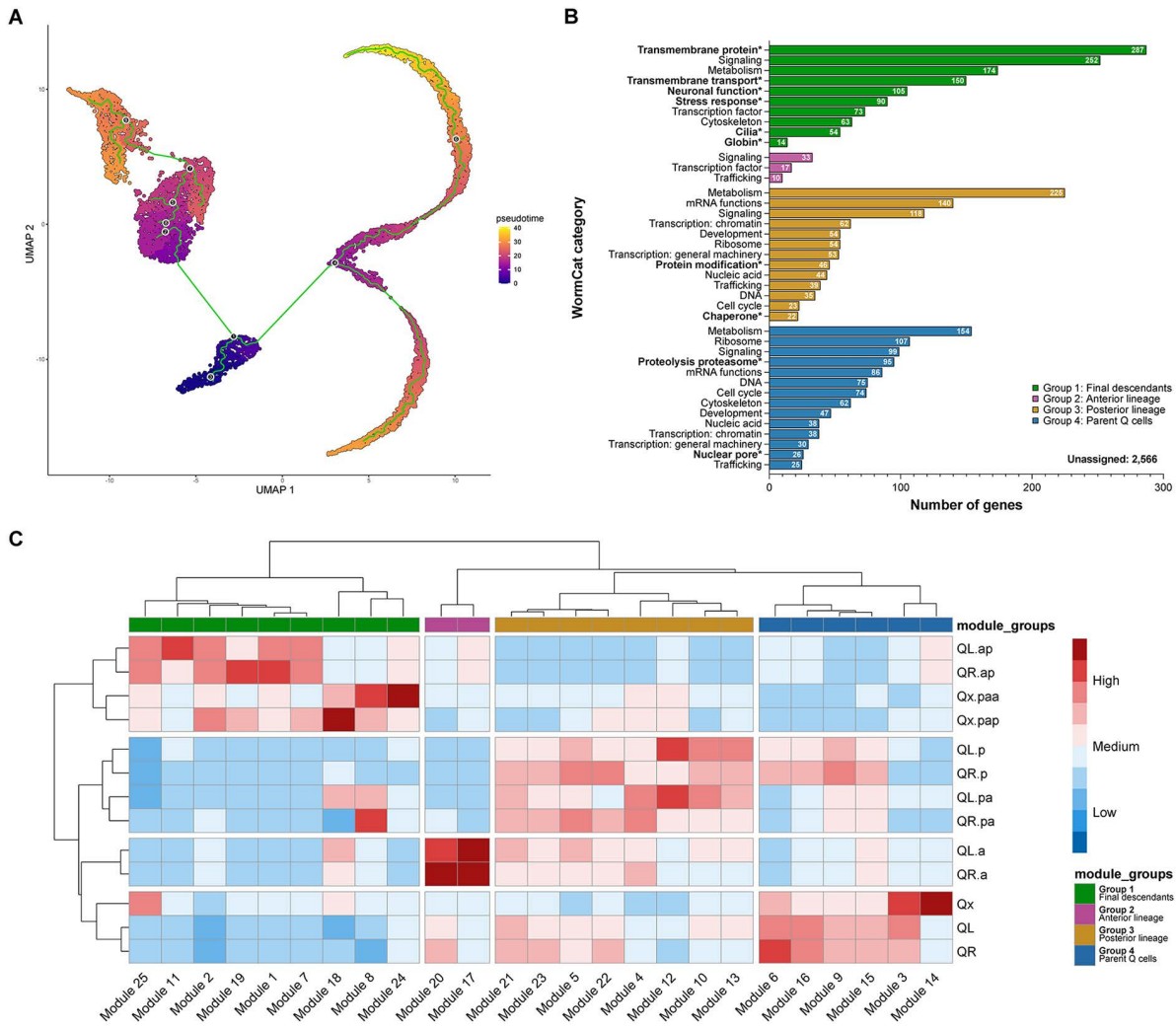

**Fig 2. Trajectory inference and gene expression dynamics. (A)** UMAP plot of the Q cell lineage showing the inferred trajectory (green line) across pseudotime. **(B)** Functional annotation of module groups obtained using WormCat. Categories in bold and marked with an * were observed only in that module group. Enrichment was determined using WormCat's standard settings (Fisher's exact test with Bonferroni correction; P<0.01), and the categories shown correspond to the broadest functional level (Category 1) provided by WormCat. **(C)** Clustered heatmap showing modules of co-regulated genes expression across different Q cell types. Module groups were established based on module clustering.

only one that had categories associated with neuronal cell fate, such as neuronal function (mainly synaptic function), cilia, and globin, consistent with the differentiation of the final descendants into neurons.

## Analysis of genes associated with initial migration and polarization of Q neuroblasts

Migration of Q neuroblast and their descendants happens in two phases controlled by different mechanisms. The second phase is controlled by Wnt-dependent signaling, but regulation of the early stage of development, when the Q neuroblast is still starting its migration to the neighboring seam cell, is Wnt-independent and not well understood [12,14]. We divided the parent Q neuroblast cluster in two (Fig 3A) so we could analyze the differences between early- and late-stage Q cells and identify major marker genes associated with Q lineage progression during the first phase of migration.

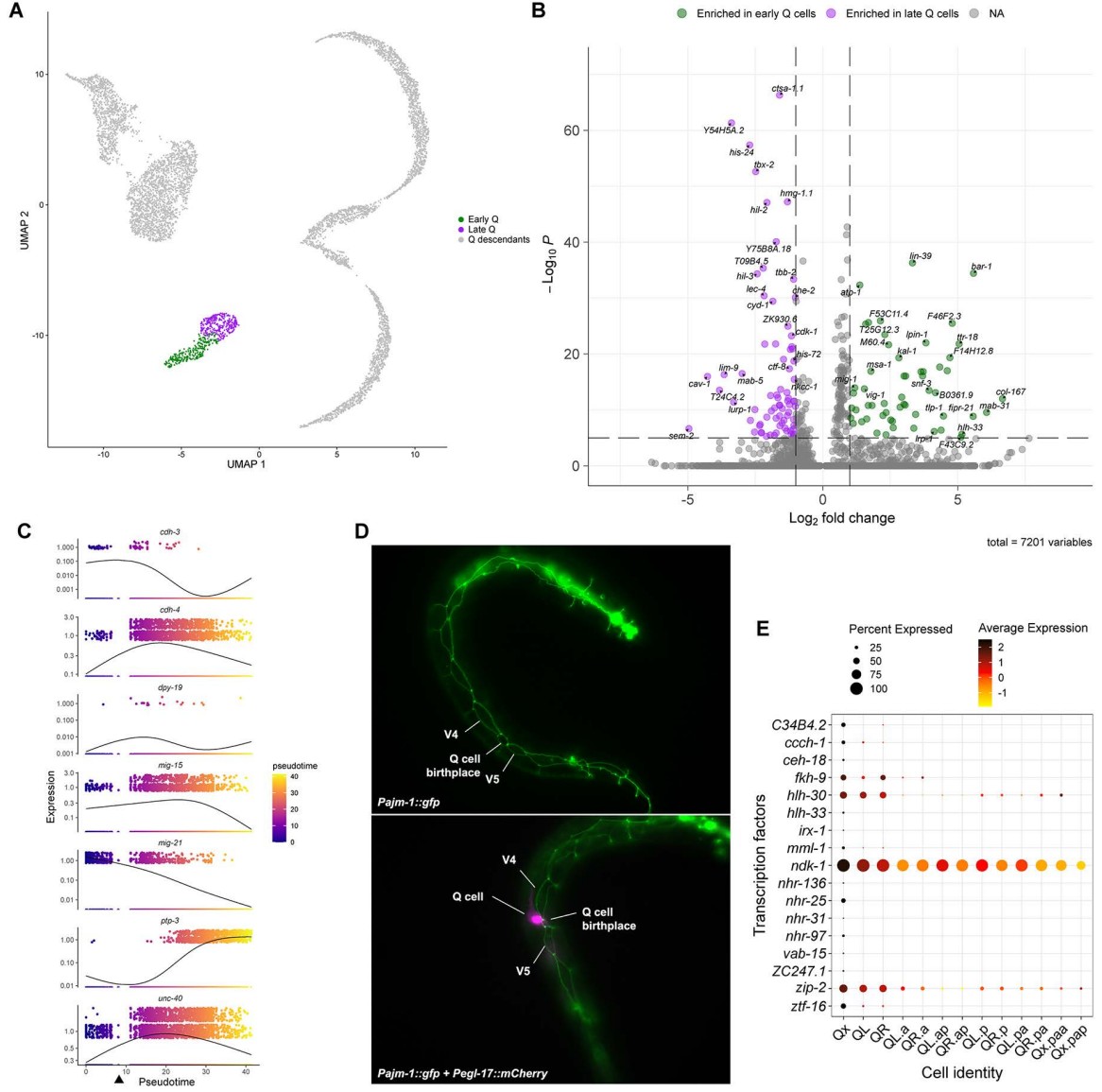

**Fig 3. Analysis of Q neuroblast initial migration. (A)** UMAP plot clustering the early and late Q cells apart. **(B)** Volcano plot showing genes differentially expressed between early and late Q cells. **(C)** Expression of genes associated with initial Q cell polarization and migration plotted in pseudotime. The arrowhead indicates the point in pseudotime in which the first cell division occurs. **(D)** Micrographs of worms during the first few hours of postembryonic development showing the expression of an apical junction marker (*ajm-1*) by Q cells during their initial migration. **(E)** Dot plot showing the average expression of transcription factors differentially expressed in early Q cells. Results were filtered to display genes expressed in at least 5% of cells in any cluster.

Among differentially expressed genes (Fig 3B; S3 File) in early Q cells, we identified several genes related or predicted to be related to the worm cuticle, such as *ram-2*, *col-3*, *col-107*, *col-167*, and *dpy-6* [51], which were downregulated as these cells transitioned to their later stage. Enrichment analysis for anatomic associations using WormEnrichr [52,53] showed that genes co-regulated in early Q cells had a high anatomic association with the worm hypodermis (S3 File), suggesting an epithelial fate for the cells at this stage. To investigate if the transition from early- to late-stage Q cells resembles EMT, we examined the first few hours of L1 development using worm strains expressing an apical

junction marker (*Pajm-1::gfp*) [54] and a Q cell marker (*Pegl-17::mCherry*) [55]. The results showed that, while Q cells were initially delimited by the apical junction marker, the protruding cell is no longer marked by it once it is migrating away from its birthplace (Fig 3D). This behavior matches the expression of *ajm-1* in Q cells (S2 Fig), that seems to still be "on" in the earliest cells in our dataset but is downregulated afterwards. We also investigated the expression of EMT-related genes and identified that *zag-1* is transiently expressed in early Q cells and subsequently turned off as the cells progress toward their later fate (S2 Fig). ZAG-1, a member of the ZEB family of TF, is known in vertebrates to regulate EMT [56], further supporting the possibility that Q cells undergo an EMT-like program during the initial phase of migration. However, ZAG-1 has also been described in *C. elegans* as a regulator of neuronal differentiation [56]. The observed re-activation of *zag-1* later in Qx.p suggests a dual role for this TF in the lineage, potentially affecting differentiation of the Qx.p lineage.

During the first phase, QL and QR will polarize in opposite directions and then undergo a short-range migration. Several genes have been associated with the initial polarization and migration, including *mig-21*, *cdh-3*, *cdh-4*, *unc-40*, *mig-15*, *ptp-3*, and *dpy-19* (Fig 3C; S3 Fig). As described above, *mig-21* expression is high in early Q cells and decreases over time. Expression of *cdh-3* followed a similar tendency of expression early in the Q neuroblasts and downregulation upon the first cell division, but at lower levels compared to *mig-21*. CDH-3 is a Fat-like cadherin required for correct Q neuroblast polarization [57]. The expression of other genes associated with Q neuroblast migration and polarization, such as *unc-40*, which encodes a netrin receptor [11,14,57], *cdh-4*, which encodes a Fat-like cadherin [15,57], and *mig-15*, which encodes a Nck-interacting kinase (NIK) [12], was not exclusive to the early Q cells as they were expressed throughout the Q cell lineage. High expression in Qx strongly confirms or suggests a cell-autonomous role for these genes in the initial migration of Q neuroblasts. The expression pattern of *ptp-3*, on the other hand, does not make it clear if the role of *ptp-3* in Q cell migration originates from a cell-autonomous or non-autonomous effect. PTP-3, a LAR receptor tyrosine phosphatase, also contributes to Q neuroblast polarization [14,58]. *ptp-3* was expressed in the Qx cluster, but at considerably low levels – its highest expression was in the Qx.ap clusters. The expression of *dpy-19* was detected in less than 0.5% of the sequenced cells, with no reliable expression pattern and no association with any cluster. DPY-19 is a C-mannosyltransferase that glycosylates thrombospondin repeats [59] and it was previously described to act cell-autonomously in Q neuroblast polarization based on rescuing of mutant phenotype upon Q lineage-specific expression of *dpy-19* in *dpy-19* mutant worms [11,57]. Due to its negligible expression in the Q cells, however, the role of DPY-19 in Q cell migration is likely non-autonomous, though its ectopic expression in Q cells may be sufficient to rescue Q cell polarization in *dpy-19* mutant worms as previously described.

Analysis of TFs expressed at the early stage identified new candidate genes not previously associated with the Q cells (Fig 3E). Some, such as *nhr-25* and *C34B4.2*, seem to be specifically enriched during the earlier stages, while others, like *ndk-1* and *zip-2,* are enriched early but continue to be expressed later. Given the limited understanding of genes regulating Q neuroblast initial migration, further investigation of TFs expressed at this early stage may reveal key regulatory mechanisms associated with this first phase of migration.

## Analysis of the L-R asymmetry during Q lineage development

The antagonistic activity of MAB-5 and LIN-39 mediates the asymmetric migration of QL and QR descendants [9,18,19,25]. Genes differentially expressed between the two cell lineages are potential downstream targets of those Hox factors. Thus, we sought to identify genes differentially expressed between the left and right-side at different stages of Q lineage development. Differentiation between left and right-side lineages was possible once we could detect expression of *mab-5* in QL (Fig 4A). Interestingly, comparison between the whole QL and QR lineages shows more genes with asymmetric upregulation on the right side compared to the left side (Fig 4B-F; S4 Fig; S4 File), which might be related to the increased distance QR descendants have to migrate along the anterior-posterior axis to get to their final positions when compared to QL.

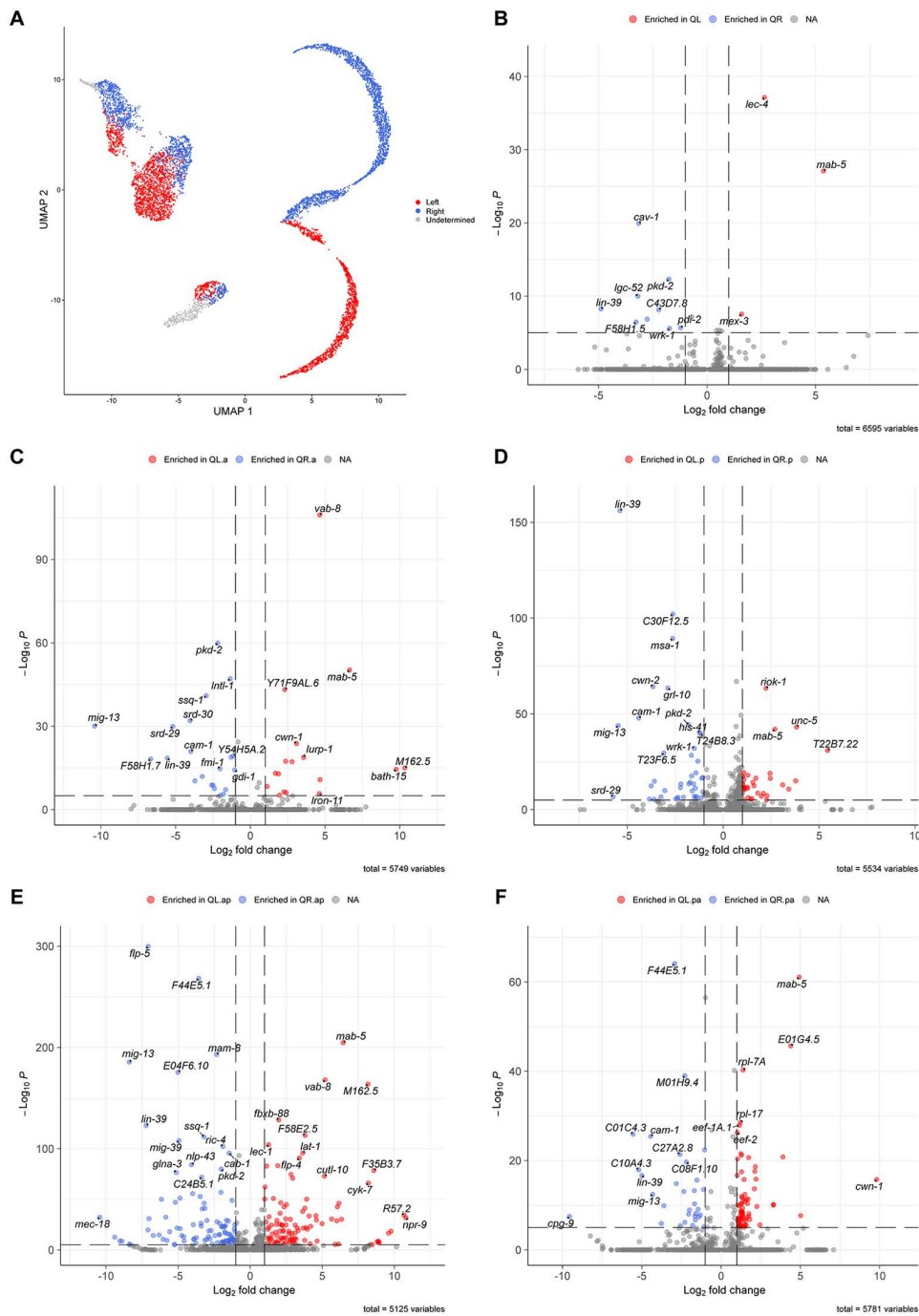

**Fig 4. Differential expression analysis of genes with L-R asymmetry. (A)** UMAP segregating cells from the left- (red) or right-side (blue) lineages. Grey cells represent clusters in which we could not differentiate left from right. **(B-F)** Volcano plots showing genes asymmetrically expressed in the parent Q cells **(B)**, the first (C) and second descendant (E) of the anterior lineage, and the first (D) and second descendant (F) of the posterior lineage.

Only a few genes had a strong differential expression between QL and QR, but their expression was not exclusive to the parent Q cells. Besides *mab-5*, *lec-4* was the gene with the most prominent expression in QL. Even though *lec-4* is also expressed in other cells within the Q lineage, it is clearly upregulated in QL compared to every other cell type (S4 Fig). LEC-4 is a galectin, a group of lectins that binds to β-galactosides, with affinity for different glycolipid-type oligosaccharides [60]. The expression of *mex-3* was also higher in QL when compared to QR (S4 Fig). MEX-3 is a KH-domain RNA-binding protein that contributes to asymmetry and cell fate specification in embryos, and to maintenance of totipotency in adult germ cells [61,62]. Expression of *mex-3* in Q cells starts before we can differentiate between QL and QR and is maintained in QL but downregulated in QR. MEX-3 may function in specification of Q cell fate during early migration and in further differentiation of QL after *mab-5* upregulation. MAB-5 might be required to sustain *mex-3* expression in QL, or LIN-39 might contribute to its downregulation in QR – or both.

Expression of the caveolin *cav-1* is enriched in QR compared to QL (S4 Fig). Caveolins are crucial for the formation of caveolae, a subtype of lipid rafts involved in numerous signaling pathways [63,64]. CAV-1 in worms is similar to mammalian caveolin-1 and cavelolin-3 [65]. Mammalian caveolin-1 has been associated with cell migration, with a role promoting cell movement and polarity [66]. Upregulation of *cav-1* in QR may contribute to the migration asymmetry of the Q lineage. However, *cav-1* is expressed in Q descendants from both sides after the first cell division. Thus, any asymmetric role would likely be restricted to the early migration and polarization phase. While *cav-1* asymmetry is maintained only in the parent QR, expression of *lgc-52*, *srd-30*, and *srd-29* starts before the first cell division but is maintained exclusively in the QR.a lineage after division. While *lgc-52* is downregulated before QR.a divides, the expression of *srd-30* and *srd-29* is maintained through the earlier stages of QR.ap migration (S4 Fig). *lgc-52* encodes a ligand-gated ion channel [67] and *srd-30* and *srd-29* encode serpentine receptors. Not much has been described about these genes in worms so far.

Differences between left- and right-side lineages become more evident in Q descendants. Among the genes with a clear association with the anterior descendant lineage, *vab-8* was the most significant in the left side. *vab-8* is upregulated early in QL.a and retains its expression for most of QL.ap trajectory (S4 Fig). This expression pattern is consistent with previous reports indicating that mutations in *vab-8* disrupt posterior migration of QL.ap, leading to anteriorly misplaced PQR [68], and it supports the Q cell-autonomous role recently suggested for this gene in Q lineage migration [69]. The left-side enrichment of *vab-8* is also consistent with previous bulk RNA sequencing data that identified *vab-8* as a potential target for MAB-5 [18]. Expression of *ssq-1* follows an anterior-specific pattern similar to *vab-8*, but on the right-side lineage, with high expression in the QR.a lineage up to late QR.ap (S4 Fig). The function of SSQ-1, a member of the sperm-specific family, class Q, is still unknown in *C. elegans* [70].

Among the genes with a clear association with the posterior descendant lineage, *T22B7.22* was the most enriched in QL.p (followed by *H37A05.4* and *unc-5*) and *Y45G5AM.5* was the most enriched in QL.pa (S4 Fig). On the right side, most of the genes enriched in QR.p against QL.p are also expressed in the anterior lineage. *grl-10* is the top gene enriched in QR.p against the rest of the dataset, though it still presents a low expression in early QL.p cells (S4 Fig). *grl-10* is a hedgehog-related gene with expression previously described in seam and rectal epithelial cells [71]. *C01C4.3* is the top gene enriched in QR.pa against the entire dataset (S4 Fig).

Some genes with L-R asymmetric expression were associated with both anterior and posterior lineages. The asymmetry was more evident on the right side, where *mig-13* and *cam-1* were the top genes with high expression exclusively in the QR lineage (S4 Fig). The importance of both genes to the anterior migration of the QR lineage has been described before [25,72,73]. On the left side, the expression of *bath-15*, predicted to be involved in the *C. elegans* ubiquitination pathway, was strongly associated with QL.a and QL.p lineages (S4 Fig). Even though the differential expression is not significant in the early Q neuroblasts due to low expression levels, the expression asymmetry in QL versus QR and the expression trend in the descendants suggests that *bath-15* expression pattern might be biologically relevant for the QL lineage starting in the Q neuroblasts.

Most markers associated with the final descendants are active in the differentiated neurons and can be identified in other datasets [29,34]. The most notable exceptions encompass transiently expressed genes that are turned off before neuron maturation, such as *M162.5* on the left side and *F54F12.2* on the right side (S4 Fig), neither detected among the genes enriched in PQR or AQR neurons, respectively, in CeNGEN L1 data [34]. Both genes encode proteins predicted to be in the cell membrane with no role described in worms yet.

## Expression pattern of genes deregulated in *mab-5* mutants

Previous bulk RNA sequencing analysis of Q neuroblasts isolated from *mab-5* loss-of-function (lof) and gain-of-function (gof) mutant worms revealed potential downstream targets of MAB-5 [18]. We analyzed the expression pattern of those genes to identify cases where L-R asymmetry in gene expression might correlate with MAB-5 regulation, while also identifying the cell types the genes are associated with.

Among the genes downregulated in either the lof or gof mutants, several presented an asymmetric pattern of expression in our dataset. Genes from the lof mutant were mostly associated with the left side, suggesting they might indeed be targets of MAB-5 (S5 Fig). For example, *M162.5* is upregulated in late QL.a and throughout QL.ap, suggesting it is important for PQR development, while expression of *sem-2* is detected in late QL and early QL.p, suggesting it is important for PVM and/or SDQL. On the other hand, L-R asymmetric genes downregulated in the gof included *lin-39* and were mostly associated with the right side, suggesting that gof changes are likely due to the QR lineage assuming a QL-like fate (S7 Fig).

Upregulated genes, however, rarely presented the L-R asymmetric pattern. In the *mab-5* lof, they were more likely to have no expression in the Q lineage at all, which could be due to non-autonomous effects of *mab-5* mutation triggering the expression of genes not usually expressed in the Q lineage (S6 Fig). The lof upregulated genes that were expressed in the Q lineage were more likely to be associated with earlier expression, which suggests that the Q cell population isolated from the lof mutant strain was more likely to have a higher concentration of cells in earlier stages than the wild-type strain. In the gof, most upregulated genes were not expressed in Q cells either, but the ones that were, had no clear association with a specific cell type or stage (S8 Fig).

## Identification of genes associated with anterior-posterior lineage asymmetry

Since anterior-posterior lineage asymmetry seems to play a more dominant role than L-R asymmetry in Q cell development, we sought to identify genetic markers linked to cell lineage progression and different cell types regardless of L-R asymmetry. We removed L-R identity from our clusters (Fig 5A) and used Seurat's FindAllMarkers function to identify marker genes associated with each cell type (S5 File). As expected, the terminally differentiated anterior descendants (Qx.ap) were characterized by a strong enrichment of neuronal markers, reflecting their postmitotic cell state. In contrast, fewer positively enriched markers were detected for transient neuroblast populations such as Qx.p (Fig 5B; S5 File; S9 Fig).

We performed differential expression analysis between only Qx.a and Qx.p cells to identify genes associated with the initial anterior-posterior lineage asymmetry, which should help identifying more markers associated with lineage progression across the anterior or posterior lineages (Fig 5C; S5 File). The main difference between the two lineages was the number of markers identified, suggesting that Qx.p functions as a transition state between Qx and Qx.pa, with only a limited set of genes uniquely expressed in these cells. The genes with the highest significance in Qx.p (*unc-86*, *F32H5.3*, *hphd-1*, and *cyd-1*) were among the top markers of this cluster in Fig 5B. Qx.a, on the other hand, displayed considerably more differentially regulated genes when compared to Qx.p, some of them also present in Qx.ap. These patterns reinforce the idea that Qx.p retains a neuroblast-like transition state, whereas Qx.a already shows signs of neuronal differentiation.

Analysis of the identified markers showed that genes previously described in Q cells seemed to have an expression pattern consistent with those expected based on described mutant phenotypes or molecular analyses. For example, we

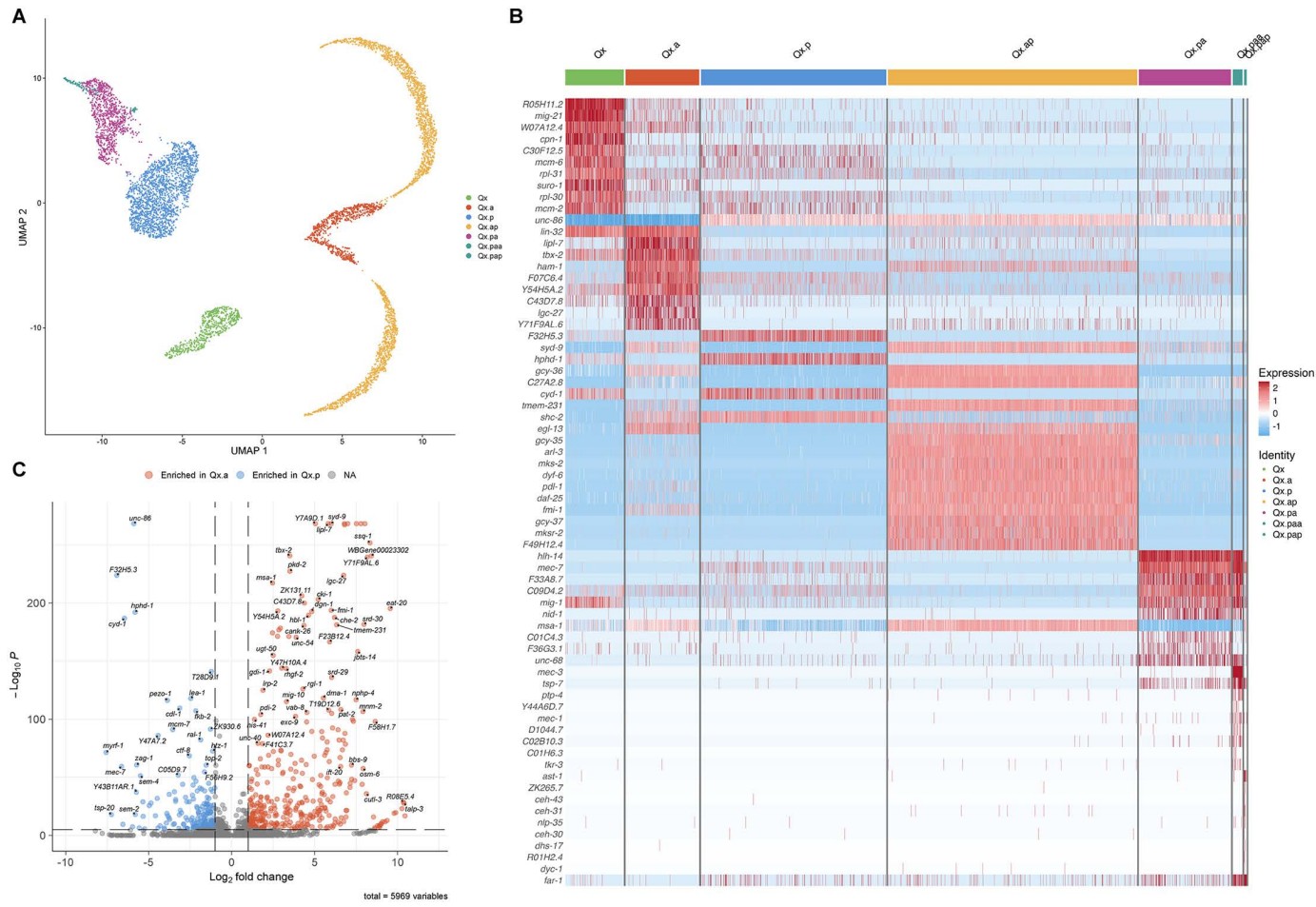

**Fig 5. Analysis of genes differentially expressed between lineages and cell types regardless of L-R asymmetry. (A)** UMAP plot showing Q cell types clustering without L-R segregation. **(B)** Heatmap showing the top 10 markers differentially expressed in each cluster. **(C)** Volcano plot showing genes differentially expressed between the first descendants of the anterior and posterior lineages.

show that the expression of *unc-86* is enriched in Qx.p compared to Qx and Qx.a (S9 Fig). UNC-86 is a POU TF important for Qx.p cell fate. In *unc-86* mutants, Qx.p repeats the division pattern of the parent Q neuroblast and will ultimately give rise to extra AQR/PQR neurons instead of AVM/PVM [41,74,75]. The expression of the proneural TF *lin-32*, on the other hand, was enriched in Qx.a against Qx.p (S9 Fig), consistent with previously reported enrichment of LIN-32::GFP in Qx.a against Qx.p [76]. LIN-32 works in parallel with HAM-1, a storkhead TF also enriched in Qx.a, to control Qx.a asymmetric cell division [76].

Most of the genes, however, have not been previously described in Q cells. Some, such as *R05H11.2*, *lipl-7*, and *F32H5.3* (S9 Fig), were not even properly studied in worms before, yet show strong expression in specific Q cell types. For those that have been studied in other cell types, existing data may offer clues about potential roles in the Q cells. For example, the basic helix-loop-helix TF *hlh-14* was enriched in Qx.pa and Qx.paa (S9 Fig). Studies have shown HLH-14 function is important for correct asymmetric cell division and descendant fate in other neuroblast lineages [77]. The expression pattern in Q cells suggests that *hlh-14* might have a role in Qx.paa/pap asymmetric division and AVM/PVM neuronal fate. It also has been reported that *hlh-14* has a mirror image expression pattern relative to *lin-32* in neuronal

lineages during embryonic development [78], a similar pattern observed in the expression of these genes in the anterior and posterior Q lineage, with *lin-32* associated with the AQR/PQR neurons and *hlh-14* associated with AVM/PVM neurons.

## Expression pattern of genes associated with the Wnt signaling pathway

No role for Wnt pathway signaling has been described in L-R asymmetry during the initial polarization and migration of the parent Q neuroblast, but the Wnt cascade triggered by the Wnt ligand EGL-20 is essential to maintain the L-R asymmetry during the migration of Q descendants. EGL-20 is produced by and spreads from a group of cells in the posterior region of the animal, forming a long-range gradient along the anterior-posterior axis [21,79,80]. While EGL-20 is crucial for posterior migration of the QL lineage, it can also influence anterior migration of QR descendants [21,26]. We analyzed the expression of different genes associated with the Wnt pathway [81] to understand how this pathway can affect asymmetry during Q cells development (Fig 6A).

The most evident L-R asymmetry was found in the expression of Wnt ligands and receptors. While no expression could be observed for *egl-20* as expected, two other Wnt ligands, *cwn-1* and *cwn-2*, were expressed in the Q cells with a L-R asymmetric pattern (Fig 6B, C). Expression of *cwn-2* starts in early QR cells and stays on throughout QR lineage development. Expression of *cwn-1* is upregulated in the QL lineage after the first division, with high expression in QL.a/ap cells but also present in QL.p/pa. A previous study described an autocrine role for CWN-1, acting in parallel to EGL-20, in inhibiting anterior migration of the QL lineage [18]. The expression pattern in our scRNAseq dataset suggests that both CWN-1 and CWN-2 might act cell-autonomously, with possible autocrine roles, in the QL and QR descendants, respectively.

The Wnt receptors with the most evident expression during Q cell lineage development include the Frizzled receptors *mig-1*, *lin-17* and *mom-5*, and the Ror ortholog *cam-1*. Among these, *mig-1* displays anterior-posterior lineage asymmetry, while *lin-17*, *mom-5*, and *cam-1* show L-R asymmetry (S10 Fig). Our data shows that *mig-1* is expressed early in the parent Q cells, with downregulation during the first phase of migration and cell division. It is upregulated again in the posterior lineage, especially in Qx.pa, in a similar fashion to previous smFISH analyses that showed high expression of *mig-1* in QR, late QR.p, and QR.pa, but with our data showing this pattern is observed on both sides [26,82]. It has been reported that *mab-5* activation requires *lin-17* [19]. It has also been reported that *lin-17* is differentially regulated by MAB-5 and that LIN-17 acts downstream of MAB-5 [69]. The expression of *lin-17* was higher in QL and QL descendants but also present at lower levels in QR and QR descendants. The expression of *cam-1*, on the other hand, was restricted to the QR lineage, as mentioned in a previous section. The expression of *mom-5* was low and seemed more associated with the right side and later stages of development.

EGL-20 triggers a canonical Wnt/β-catenin signaling in QL that culminates in BAR-1/β-catenin binding to the TCF TF POP-1 and co-activating the expression of *mab-5*, directing posterior migration of QL descendants. In the QR lineage, EGL-20 seems to influence anterior migration by non-canonical Wnt signaling, as posterior migration in the QL lineage is severely impaired in both *bar-1* and *pop-1* mutants, while no significant defect in migration might be observed in the QR lineage [9,19,21,81]. *bar-1* was expressed in the earliest cells in our dataset, with no apparent difference that would cluster QL and QR apart. It was downregulated as QL and QR started differentiating, in a clear contrast with *pop-1*, which was upregulated as *bar-1* expression decreased and remained "on" throughout the development of both QL and QR lineages (Fig 6D, E). BAR-1 is the only β-catenin with a role described in Q cells [81], but we found that it is not the only β-catenin expressed in them. Notably, both *sys-1* and *wrm-1* are upregulated as the Q cells differentiate into QL and QR and remain expressed through most of Q lineage development (Fig 6F, G). SYS-1 and WRM-1 are the two β-catenins associated with the Wnt/β-catenin asymmetry pathway, which control asymmetric cell divisions [83] such as the one that generates Qx.a and Qx.p, as well as their descendants.

## Discussion

In this work, we have used scRNA-seq to generate a comprehensive transcriptomic map of the entire Q neuroblast lineage during migration and differentiation into neurons, with a clear distinction between the Q neuroblasts and their

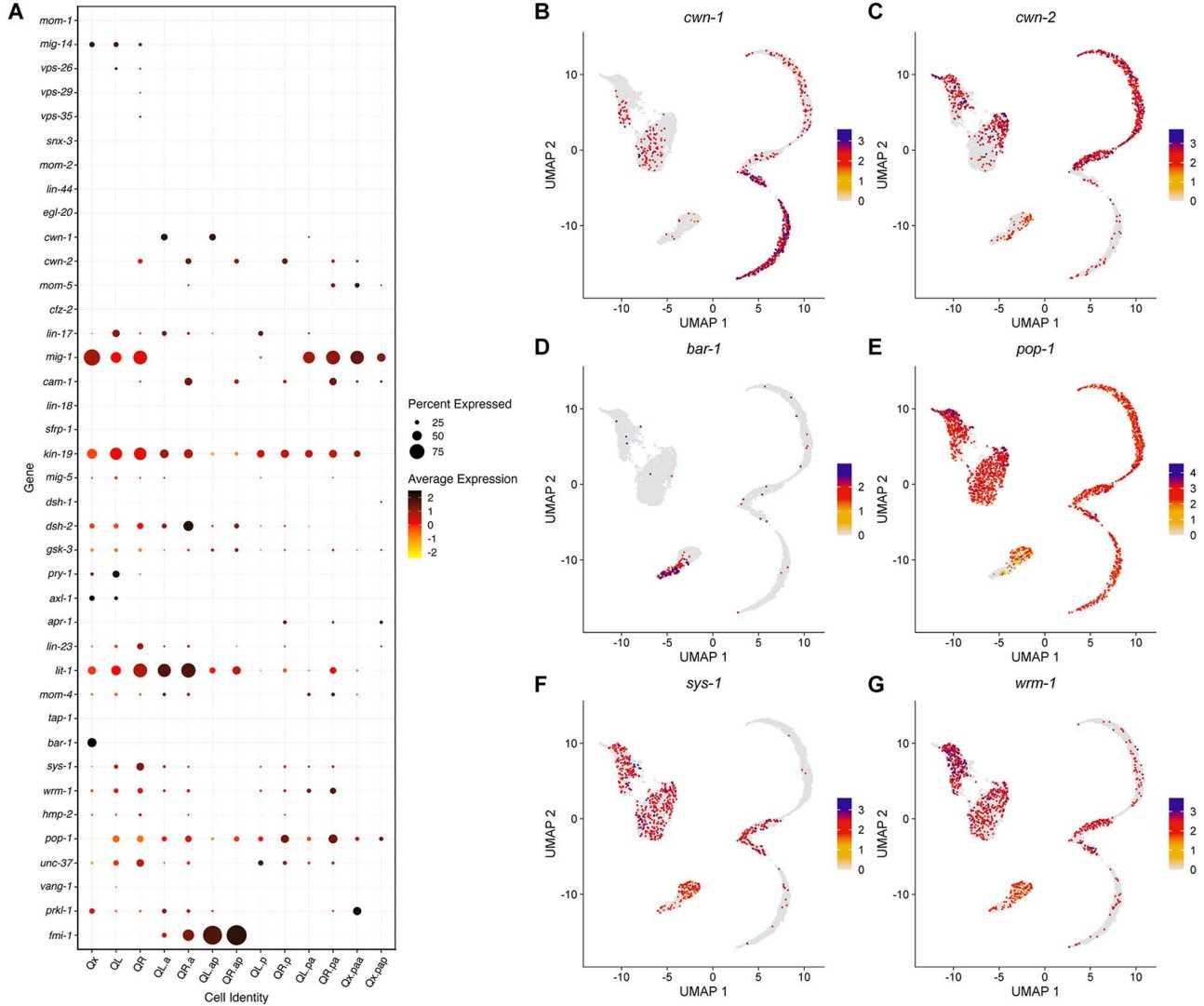

**Fig 6. Expression analysis of Wnt-related genes. (A)** Dot plot showing the average expression of Wnt-related genes in different Q cell types. Results were filtered to display genes expressed in at least 10% of cells in any cluster. **(B-G)** UMAP plots showing the expression pattern of selected Wnt-related genes.

descendants. This approach enabled the discovery of novel genes associated with the Q cell lineage, some of them unlikely to be identified through traditional genetic screens due to subtle or non-obvious phenotypes, lethality of mutation, or functional redundancy within the system. We believe our findings provide a valuable foundation for future investigations.

## The L-R asymmetry during Q cell lineage migration

Our data showed that the L-R asymmetry is not as strong as we previously expected, especially at earlier stages. Most of the time, cluster assignment had a stronger association with Q lineage progression (i.e., Qx to Qx.a and Qx.p) than with L-R asymmetry (i.e., QL.x vs QR.x), suggesting that lineage progression is wired deeper into the cells and L-R asymmetry is layered on top. The most evident L-R asymmetry was observed in the anterior lineage, as Qx.a differentiate into Qx.ap.

This is consistent with MAB-5 being a Hox terminal selector in the QL lineage [18], controlling the expression of effector genes that will differentiate the left from the right-side lineage to establish the neuronal identity of its final descendants (PQR, PVM, and SDQL).

### The first phase of Q cell migration seems to be an epithelial-mesenchymal transition

The Q neuroblast is sister to the seam cell V5 and is born from ABpxapapaa approximately one hour before hatching, starting its migration only after the larva hatches from the egg [7]. Our results suggest that the Q cell might present an epithelial cell fate by the time it is born and later, during the first phase of migration, transition to the neuroblast fate. This hypothesis was suggested by enrichment analysis showing that the gene modules expressed in the early Q cells were anatomically associated with an epithelial cell fate. Also, the transcriptomic data showed that early Q neuroblasts express low levels of *zag-1*, an ortholog of ZEB1 and ZEB2, TFs associated with induction of EMT in humans [84]. They also express genes linked to cuticle formation, a trait typical of epithelial cells such as hypodermis and seam cells, which synthesize various collagen proteins essential for the cuticle [85]. Microscopy analysis showed that the Q cells appear to be delimited by the apical junction marker AJM-1 only at their birthplace, losing marker expression as they migrate (Fig 3D).

### Autonomous expression of *cwn-1* and *cwn-2* shows new L-R asymmetry for Wnt signaling in the Q cells

We have previously shown that expression of *cwn-1* in QL descendants is regulated by MAB-5 and that CWN-1 acts as an inhibitor of anterior migration in these cells, possibly in parallel with EGL-20 [18]. Although *mab-5* gof leads to downregulation of *cwn-2* in Q cells according to bulk RNA-seq analysis [18], the molecular trigger for wild-type *cwn-2* expression in the QR lineage is still unknown. Our current data shows that expression of *cwn-2* starts early, before we can reliably differentiate between QL and QR. Given its L-R asymmetric expression pattern throughout the lineage though, it is likely that the early Q cells expressing *cwn-2* are QR, which makes it a potential marker for QR identification in earlier stages. The autonomous expression pattern of *cwn-2* suggests the potential for autocrine signaling during early stages of QR development. Given that overexpression of *egl-20* can activate *mab-5* expression in QR [21] and that binding between Wnt ligands and their receptors can be promiscuous [86,87], it is possible that autocrine CWN-2 competes with EGL-20 for binding, leading to the higher EGL-20 threshold needed for *mab-5* activation in QR. Another hypothesis is that autocrine CWN-2 signaling complements EGL-20 signaling to modulate cell response and prevent *mab-5* expression in QR, but overexpression of *egl-20* disrupts this balance.

Despite the clear L-R asymmetry we observed in our work, data from the literature show evidence of *cwn-1* and *cwn-2* acting in Q cells from both sides [16,18,26,88,89]. *egl-20* mutants display a drastic migration defect in QL descendants, which assume a QR-like fate and migrate anteriorly, but only minor defects in QR descendant migration. Single mutants for both *cwn-1* and *cwn-2* have only minor defects, but mutation of *cwn-1* and/or *cwn-2* in an *egl-20* mutant background leads to reduced anterior migration in both QL and QR descendants when compared to the *egl-20* single mutant phenotype [16,18,88]. Thus, it seems that CWN-1 and CWN-2 can act both cell-autonomously and non-autonomously in the Q cell lineage.

### Components of the Wnt canonical pathway can be symmetrically expressed despite role in L-R asymmetry

The β-catenin BAR-1 is necessary for the asymmetric expression of *mab-5* in QL, but not QR [21]. The symmetric expression pattern we observed for *bar-1*, however, prompted the question of why the QR lineage expresses *bar-1* despite the apparent absence of canonical Wnt/β-catenin signaling in the lineage. QR might undergo the same cell programming as QL and be prepared to respond to EGL-20, but another signal, maybe triggered by CWN-2, interferes and prevents QR from responding to the canonical pathway. Studies have shown that EGL-20 also localizes to QR descendants [80], but a higher threshold of EGL-20 is needed to activate the canonical Wnt/β-catenin in QR when compared to QL [21]. Given *bar-1* expression pattern, it is also possible that the canonical Wnt/β-catenin pathway is activated in both QR and QL cells

early during development, but the downstream response is different in each cell and prevents expression of *mab-5* in the QR lineage. Such differences in response might be caused by alternative Wnt signaling, like one triggered by autocrine CWN-2 discussed above. However, more experiments are necessary to validate this hypothesis, as gene expression does not necessarily correlate with protein activity.

### Expression of *sys-1* and *wrm-1* suggests a role for the Wnt/β-catenin asymmetry pathway in Q cells

The β-catenins SYS-1 and WRM-1 are associated with the Wnt/β-catenin asymmetry pathway, a pathway involved in the regulation of several asymmetric cell divisions during development and that contributes to the generation of daughter cells with different developmental fates [83,90]. Both QL and QR lineages undergo four asymmetric divisions during development, but the Wnt/β-catenin asymmetry pathway has not been described in Q cells before. Consistent expression of *sys-1* and *wrm-1* was detected in most cell clusters, with the lowest expression (sometimes completely absent) in the final descendants Qx.ap, Qx.paa, and Qx.pap, the cells in our dataset that will no longer undergo asymmetric division. This strongly suggests the presence of the Wnt/β-catenin asymmetry pathway in Q cell lineage development, with a possible overlap of effectors with the Wnt/β-catenin. Activation of the asymmetry pathway during cell division leads to the asymmetric distribution of pathway components between the daughter cells, affecting the ratio of POP-1 and SYS-1 in those cells. The difference in balance between these two key effectors will dictate if POP-1 will work alone to repress target genes or if it will form a complex with SYS-1 to activate its targets instead [91,92]. Thus, POP-1 has a crucial role in both the Wnt/β-catenin asymmetry and canonical pathways. This explains why there was no L-R asymmetry in *pop-1* expression, initially expected due to POP-1 role in activating the expression of *mab-5* in the left side only. Also, we did not observe any asymmetry in *pop-1* expression between anterior and posterior daughter cells, which has been reported in the Wnt/β-catenin asymmetry pathway [92].

## Materials and methods

### L1 larvae synchronization and dissociation

L1 larval population was synchronized and dissociated as previously described [18] with modifications that allowed recovery of fed worms that spanned the whole Q cell lineage development. Worms were transferred to the edge of the agar on 8P plates (150 mm) seeded with *Escherichia coli* NA22 and grown at 20 ˚C for 5 days. We transferred enough worms to allow crowding as the worm population moved as a single line across the plate, stimulating egg-holding so, by the end of incubation, we would have worms holding embryos that spanned more than the regular 2 h of development. If plates were near starvation (too close to the other edge of the plate) before ready for bleaching, worms were washed with sterile ddH$_2$O, pelleted and transferred to fresh 8P plates. After incubation, worms were washed from the plates with sterile ddH$_2$O, transferred to a 50 mL conical tube and allowed to settle. Excess water was removed, and 20 mL of freshly prepared hypochlorite solution (18.75 mL of ddH2O, 5 mL of Clorox bleach, 1.25 mL 10 N NaOH) was added to the tube. After 5 min of incubation with continuous homogenization, the tube was filled with M9 buffer to stop the reaction and the solution transferred to 15 mL conical tubes. The tubes were centrifuged for 2.5 min at 1,300 RCF and washed with M9 buffer five times, centrifuging for 2.5 min at 1,300 RCF and removing the supernatant each time. After the final wash, pellets were resuspended with 12 mL of cold 30% sucrose solution and tubes centrifuged for 2.5 min at 1,300 RCF. Floating embryos were collected with a sterile glass Pasteur pipette and transferred to a new 15 mL conical tube. Tubes were filled with sterile ddH$_2$O and centrifuged for 2.5 min at 1,300 RCF. The supernatant was discarded, and embryos were washed again with sterile ddH$_2$O. Embryos were transferred to 50 mL conical tubes and resuspended in 25 mL of M9. Embryo concentration was estimated by transferring 5 µL drops to a microscope slide and counting eggs on a stereomicroscope. Bacteria suspended in M9 (30-50x concentrated) were added, and tubes were incubated for 12-16h on a rotating shaker at 20˚C. Three separate populations of worms were analyzed using this workflow. To assess the robustness of Q lineage

development, one of the experiments was carried out using *Pseudomonas aeruginosa* PA14 as a food source, while the other two were performed with the standard *E. coli* OP50 strain.

Synchronized L1 larvae were transferred to 15 mL tubes and centrifuged at 1,300 RCF for 2.5 minutes. Worms were washed with sterile ddH$_2$O to remove leftover bacteria and, after centrifugation at 1,300 RCF for 2.5 min, transferred to 1.5 mL microtubes. The microtubes were filled with sterile ddH$_2$O and centrifuged at 16,000 RCF for 2 min. One additional wash with sterile ddH$_2$O was performed and supernatant was removed to leave a compact pellet. The pellet was mixed with 400 µL of SDS-DTT solution and the tube incubated at room temperature on a rotisserie shaker for 2 min. After incubation, tubes were filled with egg buffer, gently mixed, and centrifuged for 30 s at 16,000 RCF. The pellet was washed four more times, centrifuging at 16,000 RCF for 30 s. SDS-DTT treated worms were moved to a 15 mL conical tube with 3 mL of 15 mg/mL of Pronase (Sigma-Aldrich, St. Louis, MO, USA) in egg buffer. The larvae were dissociated for up to 25 min with the help of a 3 mL syringe and a 21G x 1 (0.8 mm x 25 mm) needle. The suspension containing the dissociated worms was split into 1 mL aliquots in 1.5 mL microtubes and filled with cold egg-buffer supplemented with 10% fetal bovine serum (egg-buffer-10). Cells were centrifuged at 530 RCF for 5 min at 4 ˚C. The supernatant was removed, and cells washed 2 times with 1 mL of cold egg-buffer-10, centrifuging at 530 RCF for 5 min at 4 ˚C each time. After the final wash, pellet was resuspended in 1 mL of cold egg-buffer-10 and centrifuged at 100 RCF for 2 min at 4 ˚C. The supernatant containing the cells was collected and pipetted through the top of a 35 µm filter-top tube to break up clumps. The pellet left in the microtube was washed with 1 mL of cold egg-buffer-10, centrifuged at 100 RCF for 2 min at 4 ˚C and the supernatant filtered through the top of the same 35 µm filter-top tube. The tube was kept in ice until sorting.

## FACS isolation of Q cells and single-cell RNA sequencing

FACS was performed using a BD FACS Symphony S6 equipped with a 70 µm diameter nozzle. DAPI was added to the cell suspension to a final concentration of 1 µg/mL prior to sorting to label low-quality cells. FACS gates were set to isolate cells expressing both the GFP and mCherry markers [18]. For each experiment, at least 25,000 live double-positive (GFP+; mCherry+; DAPI-) cells were sorted under the "purity" mask and into 1.5 mL microtubes containing 200 µL of cold PBS supplemented with 10% fetal bovine serum. While most cells expressing those markers are from the Q lineage, some seam cells were captured with this strategy as well (though they were later removed from the analysis). Following sorting, 10 µL of the sorted-cells solution were transferred to a new tube containing 200 µL of PBS. The sample was then re-analyzed on the BD FACS Symphony S6 using the same gating strategy employed during sorting to assess sorting efficiency and post-sort cell viability. In parallel, the remaining sorted cells were concentrated by centrifugation at 500 RCF for 12 minutes at 4 ˚C followed by removal of supernatant up to the 50 µL mark. Five microliters of concentrated cells were diluted into 20 µL of PBS for cell counting on a hemocytometer.

The single-cell RNA library was prepared following the 10X Genomics protocol for single-cell capture using the Chromium Next GEM Single Cell 3' Reagent Kit v3.1 (PN-1000269) and the 10X Genomics Chromium iX system, with a targeted cell recovery of 5,000 or 10,000 cells. The libraries were sequenced using the Illumina NextSeq2000 System with 28/90 bp paired end reads. The raw sequencing data is available at SRA (BioProject PRJNA1300790).

## Data processing and analysis

Raw FASTQ data from the three different experiments were processed with CellRanger v8.0.0 [35] as implemented in the nf-core [93] pipeline scrnaseq v2.6.0 [94], run with Nextflow v23.04.3 [95]. Reads were aligned to the WBcel235 assembly of the *C. elegans* genome (GCA_000002985.3) using the Ensembl v110 annotation [96]. The assembly FASTA and the annotation GTF were modified to include the mCherry and GFP sequences (S6 File) used to select worms for sequencing.

Analysis of the CellRanger results was performed using R v4.4.3 [97]. A rendered R notebook containing all code and analysis results is provided as a supplementary file (S7 File). The data for each sample was processed individually prior to

integration of the three samples. The counts matrices were assessed for empty droplets with DropletUtils v1.26.0 [98,99] and for cell-free mRNA contamination with SoupX v1.6.2 [100]. The filtered counts were then analyzed with Seurat v5.3.0 [36]. Genes were included if they were detected in at least 3 cells, and cells were included if they contained counts for at least 100 genes. The data were additionally filtered based on count estimates (100<nFeature_RNA<1600), total RNA detected (100<nCount_RNA<4000), as well as the percent of mitochondrial gene expression (<5%). Clustering was performed using the Seurat functions FindNeighbors and FindClusters. Dimensionality reduction for cluster visualization was performed with RunUMAP (see S7 File for full workflow and parameters) [37,38] and cell clusters were identified by expression of known marker genes, including, but not limited to, *mig-21, egl-13, gcy-32, mec-7, mec-3, lad-2, mab-5, lin-39*, and *ajm-1*. Non-Q cells (mostly from the seam cell lineage) were filtered out and further cell-type specific quality control was performed on each cluster due to differences in total RNA detected between parent and daughter Q cells (Qx: 1300<nCount_RNA<4000; Qx.a, Qx.ap, Qx.p, Qx.pa: 700<nCount_RNA<1900; Qx.paa, Qx.pap: 550<nCount_RNA<1300).

The data from all three samples were integrated using the RPCA method, with the first 11 principal components, and cells were clustered and visualized using the same Seurat workflow described above. Different clusters containing the same cell type were merged and annotated. Due to the small differences between L-R, data were exported to Loupe Browser v8.1.2 (10x Genomics) for manual annotation of left- and right-side cells based on *mab-5* and *lin-39* expression when clustering did not assign them automatically. The manual annotation was then imported and incorporated into the Seurat object. Trajectory inference and pseudotime analysis were performed using monocle3 v1.3.7 [46–49].

## Gene expression analysis

Differential expression analysis was performed using Seurat v5.3.0 [36]. The FindAllMarkers function was used to identify markers associated with each cell type, regardless of L-R asymmetry, and to compare marker genes across different experiments. The FindMarkers function and EnhancedVolcano [101] were used to identify and plot, respectively, genes differentially expressed between specific cell types. Significance thresholds for volcano plots were defined as an adjusted $p$-value $< 1 \times 10^{-6}$ and an absolute $\log_2$ fold change $> 1$. Expression patterns of different gene sets were visualized using standard Seurat plots. When no functional information was available in the literature, predicted gene annotations for differentially expressed genes were retrieved from WormBase (WS297 release) [102].

Gene expression dynamics were analyzed using Monocle3 v1.3.7 [46–49], applied to a Seurat object. The graph_test function was used to identify dynamically expressed genes, which were clustered into co-expression modules using find_gene_modules. WormCat 2.0 [50] was used to assign genes within each module group (S2 File) to *C. elegans* functional categories using its fixed genome-scale annotation background, and the resulting enrichment statistics were used descriptively to summarize functional programs associated with each module. WormEnrichr [52,53] was used to perform anatomic association analysis for genes in co-expression modules associated with early Q cells. The list of *C. elegans* TF genes used in our analysis was retrieved from the AnimalTFDB 4.0 database [103].

## Microscopy of early stage L1s

The transgenes *jcIs1[Pajm-1::gfp]* [54] and *lqIs274[Pegl-17::mCherry]* [55] were used to visualize apical junctions and Q cells, respectively, during the initial stages of Q cell migration. L1 larvae were synchronized for imaging the first hours of postembryonic development by washing NGM plates with M9 to remove larvae and adults and incubating the plate with the leftover eggs at 20 ˚C for 1–2 h. After incubation, hatched L1s were washed with M9 and transferred to a microtube. The tube was centrifuged at 5,000 RCF for 1 min and the pelleted larvae mounted on 2% agarose pads containing 5mM sodium azide. Images were obtained on a Leica DM5500 microscope using a QImaging Retiga-EXi camera and the 100x oil immersion objective. Captured images were processed using FIJI.

## Supporting information

**S1 Table. Summary of UMI, Gene, and Cell counts across different cell types.**
(XLSX)

**S1 Fig. UMAP plots showing (A-H) the expression pattern of marker genes used to annotate the Q cell lineage and (I) cell clustering by experiment.**
(TIF)

**S2 Fig. UMAP plots showing the expression pattern of genes associated with initial Q cell fate.**
(TIFF)

**S3 Fig. UMAP plots showing the expression pattern of genes associated early Q polarization and migration.**
(TIFF)

**S4 Fig. UMAP plots showing the expression pattern of genes with L-R asymmetry.**
(TIFF)

**S5 Fig. Expression pattern of genes downregulated in *mab-5* loss-of-function mutants.**
(TIF)

**S6 Fig. Expression pattern of genes upregulated in *mab-5* loss-of-function mutants.**
(TIF)

**S7 Fig. Expression pattern of genes downregulated in *mab-5* gain-of-function mutants.**
(TIF)

**S8 Fig. Expression pattern of genes downregulated in *mab-5* gain-of-function mutants.**
(TIF)

**S9 Fig. Expression patterns of lineage-associated genes linked to the anterior and posterior lineages.** (A) Heat-map showing the top 10 markers for the clusters displayed in Fig 5B, presented here with left-right segregation. (B) UMAP plots showing the expression patterns of selected marker genes across the Q lineage.
(TIF)

**S10 Fig. UMAP plots showing the expression pattern of selected Wnt-related genes.**
(TIFF)

**S1 File. Genes differentially expressed across experiments.**
(XLSX)

**S2 File. Co-regulated gene modules and WormCat analysis.**
(XLSX)

**S3 File. Genes differentially expressed and WormEnrichr analysis in early Q cells.**
(XLSX)

**S4 File. Genes differentially expressed between left- and right-side lineages.**
(XLSX)

**S5 File. Markers genes associated with each cell type regardless of L-R asymmetry.**
(XLSX)

**S6 File. FASTA sequences of sorting markers.**
(TXT)

**S7 File. Rendered R notebook providing the code used for all data analyses and the resulting output.**
(DOCX)

## Acknowledgments

We thank the University of Kansas Center for Genomics for supporting this study and members of the Lundquist and Ackley labs for their valuable discussions. The research reported here was made possible in part by the services of the following University of Kansas core laboratories and services: the Flow Cytometry Core; the Genome Sequencing Core; the HPC facilities operated by the Center for Research Computing; and the Genomic Data Science Core.

## Author contributions

**Conceptualization:** Felipe L. Teixeira, Erik A. Lundquist.

**Data curation:** Felipe L. Teixeira, Brian Sanderson.

**Formal analysis:** Felipe L. Teixeira, Brian Sanderson.

**Funding acquisition:** Felipe L. Teixeira, Erik A. Lundquist.

**Investigation:** Felipe L. Teixeira, Jennifer L. Hackett, Erik A. Lundquist.

**Methodology:** Felipe L. Teixeira, Brian Sanderson, Jennifer L. Hackett.

**Project administration:** Felipe L. Teixeira, Erik A. Lundquist.

**Resources:** Jennifer L. Hackett, Erik A. Lundquist.

**Supervision:** Erik A. Lundquist.

**Validation:** Felipe L. Teixeira, Brian Sanderson, Jennifer L. Hackett, Erik A. Lundquist.

**Visualization:** Felipe L. Teixeira, Brian Sanderson.

**Writing – original draft:** Felipe L. Teixeira.

**Writing – review & editing:** Felipe L. Teixeira, Brian Sanderson, Jennifer L. Hackett, Erik A. Lundquist.

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
