## [Decision Letter · Decision Letter 0]

11 Nov 2025

*C. elegans*

Dear Dr. Teixeira,

Thank you for submitting your manuscript to PLOS ONE. After careful consideration, we feel that it has merit but does not fully meet PLOS ONE’s publication criteria as it currently stands. Therefore, we invite you to submit a revised version of the manuscript that addresses the points raised during the review process.

We look forward to receiving your revised manuscript.

Kind regards,

Amit Singh, PhD

Academic Editor

PLOS ONE

Journal Requirements:

2. Please note that PLOS One has specific guidelines on code sharing for submissions in which author-generated code underpins the findings in the manuscript. In these cases, we expect all author-generated code to be made available without restrictions upon publication of the work.

Please review our guidelines at https://journals.plos.org/plosone/s/materials-and-software-sharing#loc-sharing-code and ensure that your code is shared in a way that follows best practice and facilitates reproducibility and reuse.

4. Please note that funding information should not appear in the Acknowledgments section or other areas of your manuscript. We will only publish funding information present in the Funding Statement section of the online submission form. Please remove any funding-related text from the manuscript.

**Additional Editor Comments:**

Please make the changes requested by the reviewer. Also, please ensure that citations are complete.

Reviewers' comments:

Reviewer's Responses to Questions

**Comments to the Author**

1. Is the manuscript technically sound, and do the data support the conclusions?

Reviewer #1: Yes

2. Has the statistical analysis been performed appropriately and rigorously?

Reviewer #1: Yes

3. Have the authors made all data underlying the findings in their manuscript fully available?

Reviewer #1: Yes

4. Is the manuscript presented in an intelligible fashion and written in standard English?

Reviewer #1: Yes

Reviewer #1: This manuscript describes a dataset of single-cell RNA-seq for cells from the C. elegans Q lineage. The Q lineage is a heavily studied model for several fundamental processes including left-right symmetry breaking, cell migration and temporal control, but targeted lineage-resolved genome-wide expression data were not previously available, and this lineage is minimally (if at all) covered in large-scale single cell atlases which have largely avoided the L1 larval stage when the Q divisions occur. Overall the dataset appears to be of high quality, with known markers of specific lineages robustly differentially expressed, and standard approaches to annotation and analysis are used to provide various candidate regulators of the different aspects of Q lineage biology. The description of the differentially expressed genes and pathways is substantial, while this makes it hard to read, it will make the text useful for people wanting a comprehensive description of Q lineage-specific expression. In addition, the data are available for others to do their own analyses. Based on all this, I recommend publication as is, although I provide a few minor suggestions for the authors to consider at their discretion.

1.

If the authors were open to additional analyses, it could be useful to compare the early Q cells to seam cells (the authors report that seam cells are present in the data and were filtered out during QC). Some of the TFs mentioned as potential regulators of early aspects of Q biology like the “EMT-like” delamination (lines 264-269) such as nhr-25 are broadly expressed across epidermal cells. Finding TFs enriched in Q relative to seam cells might help predict those more likely to be instructive for the Q-specific cell behaviors.

2.

The annotations in Figure 1C are reasonable but it is maybe worth acknowledging in the text that the precise temporal boundaries (such as between Q and Q.a or Q.p) are inferred from expression differences so could be temporally offset from the true division time by a small amount.

3.

Fig 5B – this is a nice display, and it could be useful to include a version as supplement that separated lineages by LR as well (showing markers identified with the higher resolution lineage annotations) to more clearly show how LR and lineage(a/p) differences relate to each other

4.

Line 426 – could rephrase to more clearly indicate EGL-20 is a Wnt ligand

5.

Lines 118-128 – I suggest reducing the discussion of high mitochondrial cells being Qx.aa an Qx.pp given this is uncertain and maybe not of major importance to the paper

6.

Fig. 2 – If possible I suggest giving more descriptive names to the modules (Group 3 could be “Q.a”, “Group 4” could be “Q” etc) for clarity.

7.

Fig 2B (wormcat enrichments) should ideally describe in the legend what criteria were used for enrichment (P value cutoff, fold enrichment, etc)

**Do you want your identity to be public for this peer review?** For information about this choice, including consent withdrawal, please see our Privacy Policy

Reviewer #1: No

---

## [Author Response · Author response to Decision Letter 1]

21 Nov 2025

A detailed point-by-point response to the reviewer and editor comments is included as a separate document in the submission.

---

## [Decision Letter · Decision Letter 1]

26 Dec 2025

*C. elegans*

Dear Dr. Teixeira,

Thank you for submitting your manuscript to PLOS ONE. After careful consideration, we feel that it has merit but does not fully meet PLOS ONE’s publication criteria as it currently stands. Therefore, we invite you to submit a revised version of the manuscript that addresses the points raised during the review process.

We look forward to receiving your revised manuscript.

Kind regards,

Amit Singh, PhD

Academic Editor

PLOS One

Journal Requirements:

Additional Editor Comments:

I recommend minor revision based on concerns of one of the reviewer.

Reviewers' comments:

Reviewer's Responses to Questions

**Comments to the Author**

Reviewer #2: (No Response)

2. Is the manuscript technically sound, and do the data support the conclusions?

Reviewer #2: Yes

3. Has the statistical analysis been performed appropriately and rigorously?

Reviewer #2: Yes

4. Have the authors made all data underlying the findings in their manuscript fully available?

Reviewer #2: Yes

5. Is the manuscript presented in an intelligible fashion and written in standard English?

Reviewer #2: Yes

Reviewer #2: In this manuscript, Teixeira and colleagues present a comprehensive single-cell RNA sequencing dataset of the neuronal progenitors in the Q neuroblast lineage of C. elegans. This lineage has been used to study patterns of cellular migration and differentiation for many years, and this dataset will be useful for generating new hypotheses and future experiments. The authors use standard analysis pipelines and procedures, and the conclusions are largely supported by the data. The authors have deposited the raw sequencing data at the NCBI SRA and provided analysis code to enhance reproducibility. While the data appear to be of high quality and useful, there are several instances where the description of experimental and analytical methods is missing or needs to be clarified. In addition, there are several grammatical and typographical errors that should be corrected. Particular concerns and instances are mentioned below.

Concerns:

1. This dataset was comprised of three separate experiments. From the entry at the SRA archive of the raw files and the R notebook, it became clear that one of the three samples was from worms fed PA14, while the other two were fed OP50. PA14 is toxic to C. elegans. The use of different bacterial food sources for one of the samples needs to be explicitly addressed in the methods. Ideally, differential expression analysis should be performed between food sources within each cell type to identify possible gene expression differences resulting from food source that could confound the results, especially if some cell types preferentially arise from this sample. If no differences in gene expression are seen, this can be noted. At the very least, the possibility of effects should be noted and the rationale for different food sources justified.

2. The fact that three samples were used and integrated should be mentioned much earlier in the methods.

3. I recommend the authors pursue subsetting the data to see if they can more clearly resolve L/R cells in the Q cells and Q.p lineages. This is a common approach in scRNA-seq analysis. In particular, rerunning dimensionality reduction (PCA and UMAP) on the cells without the Q.a lineages may allow for more distinct separation of L/R populations in the remaining lineages.

4. No methods are provided or context given to the identification of the “co-expressed gene modules” described beginning on line 187. Therefore, it is difficult to interpret the meaning and conclusions of this analysis. Please provide more detail about how these modules were identified.

5. The authors mention a “post-sort check” on lines 122-123 that “indicated high viability of sorted cells,” but there is no direct description of this “check” in the methods. It could be part of the cell counting on the hemocytometer prior to 10X processing, but this should be clarified. Please describe what criteria were used to assess viability and health.

6. Lines 380-383. The issue of markers associated with Qx.p being depleted can easily be remedied by changing the parameters of the FindAllMarkers function, setting the parameter “only.pos” to TRUE. If left as false, it will detect both enrichment (positive log2fold change) and depletion (negative log2fold change). This can be used to identify positive markers. Qx.p will likely have fewer than 10 top markers.

7. The analysis described in 372-383 uses a one group vs all others approach. It may not be surprising, then, that the differentiated neuron group Qx.ap (consisting of AQR/PQR) shows a higher number of differentially expressed genes when compared to all the other groups, which are almost entirely transient neuroblast populations. I think this section can be simplified by highlight only the positive markers for each lineal group vs others (see comment above) and moving directly into the analysis between Qx.a and Qx.p featured in the next paragraph.

8. The volcano plots indicate thresholds for both log2 fold change and p-values were used in determining differential gene expression. The exact thresholds used need to be reported explicitly in the Methods.

9. The set of genes used as the background for the WormCat enrichment comparisons should be reported. The proper set of genes for this comparison should be limited to genes that are detected in the dataset, rather than the entire set of genes with WormCat annotations.

10. Lines 401-402. The anterior lineage does differentiate earlier than the posterior lineage, as known from the initial lineage tracing from Sulston and colleagues.

11. Figures S5, S6, S7, and S8 might be more easily interpretable with respect to L/R differences if the cell types on the x-axis were grouped by L/R and then sorted by lineal generation.

12. The authors appear to use zag-1 expression as an indicator of epithelial identity, but its highest expression is in Q.p progenitors, which arise after the purported EMT in this lineage. There is no explanation of this with respect to the argument of EMT.

13. The authors mention the inconsistency of egl-17 expression as a reason for low counts of some postmitotic neurons and reference their own single-cell data. It would be more convincing as a reason for the low sampling to see images of the reporter strain in the various Q lineage progenitors, as it is the levels of GFP from the reporter that directly affect sorting efficiency.

14. In abstract: lines 23-25: The use of the word “mechanisms” suggests experiments were done to test a role for novel Wnt-related genes in this manuscript. This might be more accurate if changed from “mechanisms” to “expression.”

15. There are multiple instances where the word “on” should be changed to “in.” See Lines 42, 91, 243, and possibly others.

16. Lines 112-113: “The final descendants of the posterior Q cell lineage, Qx.paa and Qx.pap, were the smaller population in our dataset” – change to “smallest populations”

17. Line 657 – “Clustering was performed with UMAP” UMAP is a dimensionality reduction algorithm, not a clustering algorithm. The UMAP algorithm does not assign cells into discrete clusters. The clustering algorithm used (or function in Seurat) should be explicitly described.

18. Line 137 – “transcription factor (TFs)” should be “transcription factor (TF)”

19. Lines 202-203 “goblin” should be “globin.”

20. Lines 203-203 “confirming the differentiation” would be better as “consistent with the differentiation”

21. Line 302 – should read “similar to mammalian caveolin-1 and caveolin-3”

22. Lines 325-326 change “expressive” to “enriched”

23. Line 353 – This sentence made it sound like the authors were looking at genes that were commonly downregulated by both lof and gof mab-5 mutations, but this is not the case. This first sentence should be reworded to clarify.

24. Line 381 – “top 10 genes expressed” should be “top 10 genes enriched” or “top 10 genes differentially expressed”

25. Line 395 – “seemed previously” – maybe this should be “seen previously”

26. Sentence beginning at line 469 is a run-on and should split into separate sentences.

27. Multiple issues with plural vs singular agreement (lines 476, 483, possibly elsewhere). Please check for and correct these instances.

28. Line 491 “cells segregation into clusters” reads awkwardly.

**Do you want your identity to be public for this peer review?** For information about this choice, including consent withdrawal, please see our Privacy Policy

Reviewer #2: No

---

## [Author Response · Author response to Decision Letter 2]

16 Jan 2026

As requested, responses to all reviewer and editorial comments are provided in the uploaded response letter.

---

## [Editor Report · Decision Letter 2]

11 Feb 2026

Single-cell transcriptomic profiling of *C. elegans* Q neuroblast lineage during migration and differentiation

PONE-D-25-57136R2

Dear Dr. Teixeira,

We’re pleased to inform you that your manuscript has been judged scientifically suitable for publication and will be formally accepted for publication once it meets all outstanding technical requirements.

Kind regards,

Amit Singh, PhD

Academic Editor

PLOS One

Additional Editor Comments (optional):

I recommend acceptance of the manuscript
---

## [Editor Report · Acceptance letter]

PONE-D-25-57136R2

PLOS One

Dear Dr. Teixeira,

I'm pleased to inform you that your manuscript has been deemed suitable for publication in PLOS One. Congratulations! Your manuscript is now being handed over to our production team.

Kind regards,

on behalf of

Dr. Amit Singh

Academic Editor

PLOS One